# Exploring sexual contact networks by analyzing a nationwide commercial-sex review website

**Hiromu Ito**[1], **Keiko Shigeta**[1], **Taro Yamamoto**[1], **Satoru Morita**[2]*

**1** Department of International Health and Medical Anthropology, Institute of Tropical Medicine, Nagasaki University, Nagasaki, Japan, **2** Department of Mathematical and Systems Engineering, Shizuoka University, Hamamatsu, Shizuoka, Japan

* morita.satoru@shizuoka.ac.jp

**Data Availability Statement:** All relevant data are within the manuscript and its Supporting Information files.

**Funding:** This work was partially supported by the Japan Society for the Promotion of Science (JSPS)

## Abstract

Understanding the structure of human sexual contact networks is vital in a broad range of disciplines, including sociology, biology, public health, and anthropology. However, sexual contact networks are yet to be understood because technical and privacy issues make it difficult to conduct accurate, large-scale surveys. In this study, we surveyed data openly available on one of the largest adult entertainment websites in Japan, where male clients (MCs) can write online customer reviews of female commercial sex workers (FCSWs). In particular, our investigation focused on a type of establishment called "soapland," the only type of sex industry in Japan where sexual intercourse is publicly permitted. Soaplands are scattered throughout Japan, and the study website covers approximately 66% of them. Using such a vast amount of data on a nationwide scale, we clarified the network structure of commercial sex, characterized by small-world, scale-free, and disassortative mating properties. To study geographical characteristics, we compared the resulting network with three different artificially generated networks via the random rewiring of links. Moreover, we considered a simple epidemic model on the resulting network, and investigated whether it would be more effective to provide infection control measures to FCSWs or MCs. We determined that active FCSWs constitute an important pathway of infection propagation in commercial sex networks, but MCs also play an essential role as weak ties.

## Introduction

Social network analysis, which focuses on social ties rather than individual characteristics [1,2], is an effective tool for understanding social structures. For example, collaboration networks, such as co-starring relationships in movies and coauthorship relationships among researchers, have been well-studied for more than 20 years [3,4]. Recently, online friendships have been investigated using large-scale data from social networking services, e.g., Facebook and Twitter [5]. Sexual contact is a direct and fundamental contact in human society, and its study is of great importance in a wide variety of fields [6]. However, technical and privacy issues have made it difficult to conduct comprehensive research on sexual contact networks.

KAKENHI (nos. 17H04731, 19KK0262, 21H01575 to H.I.; JP19H05731 to T.Y.; and 18K03453, 21K03387 to S.M.) and the Joint Usage / Research Center on Tropical Disease, Institute of Tropical Medicine, Nagasaki University (2022-Seeds-02). The funders have/had no role in the study design, data collection and analysis, decision to publish, or preparation of the manuscript.

**Competing interests:** The authors have declared that no competing interests exist.

Several studies have investigated the number of partners engaged in sexual contact with a respondent without revealing partner details [7–10]. While it has traditionally been extremely difficult to obtain information on "who had sexual contact with whom," recently studies have been conducted using sex industry websites that publish information that provides a basis for assuming that sexual contact has occurred [11–16].

In this paper, we surveyed a Japanese commercial adult entertainment website that allows male clients (MCs) to write reviews and describe their experiences with female commercial sex workers (FCSWs) registered there. Users (MCs) write reviews under their registered names to ensure credibility. Such reviews can be used as data on actual sexual contacts between FCSWs and MCs [17]. We constructed a review network between MCs and FCSWs and analyzed its properties, focusing only on FCSWs working in Japanese brothels called "soaplands" that have existed since the 1960s. Sex establishments other than soaplands are prohibited from offering sexual intercourse in exchange for money (for more information on soaplands, see S1 Appendix). FCSWs can work legally at a licensed soapland, and they can engage in sex work while being protected from violence and various problems caused by the MCs by the soapland to which they belong, unlike street sex workers.

In 2021, there were 1,185 soaplands regulated by the National Police Agency. Soaplands were scattered throughout Japan, and the study website covered 66% of them (Fig 1A). We focused on reviews that were posted on the target website and were available to anyone online on April 4, 2021. Here, 96.4% of the reviews on the site at that time were written after the year 2020 (S1 Fig). We considered a sexual contact had occurred between the review writers (MCs) and review receivers (FCSWs) and built sexual contact networks they composed. This nationwide sexual contact network we obtained is the largest to date: it is much larger and more exhaustive than those previously studied in Brazil [11] and the UK [16]. We performed network analysis and compared the empirical network to three randomized artificial networks to see the tendency of each MC to limit his contacts with FCSWs to specific prefecture (the largest administrative division in Japan, see Fig 1A for the spatial division) and establishment. Soaplands are licensed by ordinances of each prefecture and are monitored by the prefecture police. Moreover, we examined how the network structure changes when top FCSWs or MCs are removed, considering a simple epidemiological model. This study provides important baseline material for the structural characteristics of large-scale commercial sex networks.

## Results

To clarify the geographical appearance of the review network, we show coarse-grained visualizations focusing on prefectures (Fig 1B) or establishments (Fig 1C). Some prefectures have banned soaplands. Fig 1B shows weighted links between prefectures, which are defined as the number of MCs who wrote reviews of FCSWs in the both prefectures. In other words, the thickness of the line between prefectures indicates how often MCs used establishments across prefectures. At the establishment level, the network is connected (Fig 1C), but at the individual level, there are many small, connected components in addition to the giant connected component (Fig 1D and Table 1). The enlarged panel in Fig 1D shows that there are many patterns that link one FCSW to many MCs and vice versa. We focus on the bipartite network consisting of the individual connections between FCSWs and MCs (Fig 1D), and there are no links between the same groups (left panel in Fig 2A). The properties of the bipartite network are summarized in Table 1. To clarify the characteristics of the observed network, we consider three types of artificial networks: Type I, randomly rewired networks preserving the degrees of all nodes; Type II, randomly rewired networks without changing the prefectures and degree; and Type III, randomly rewired networks without changing either the establishments or the

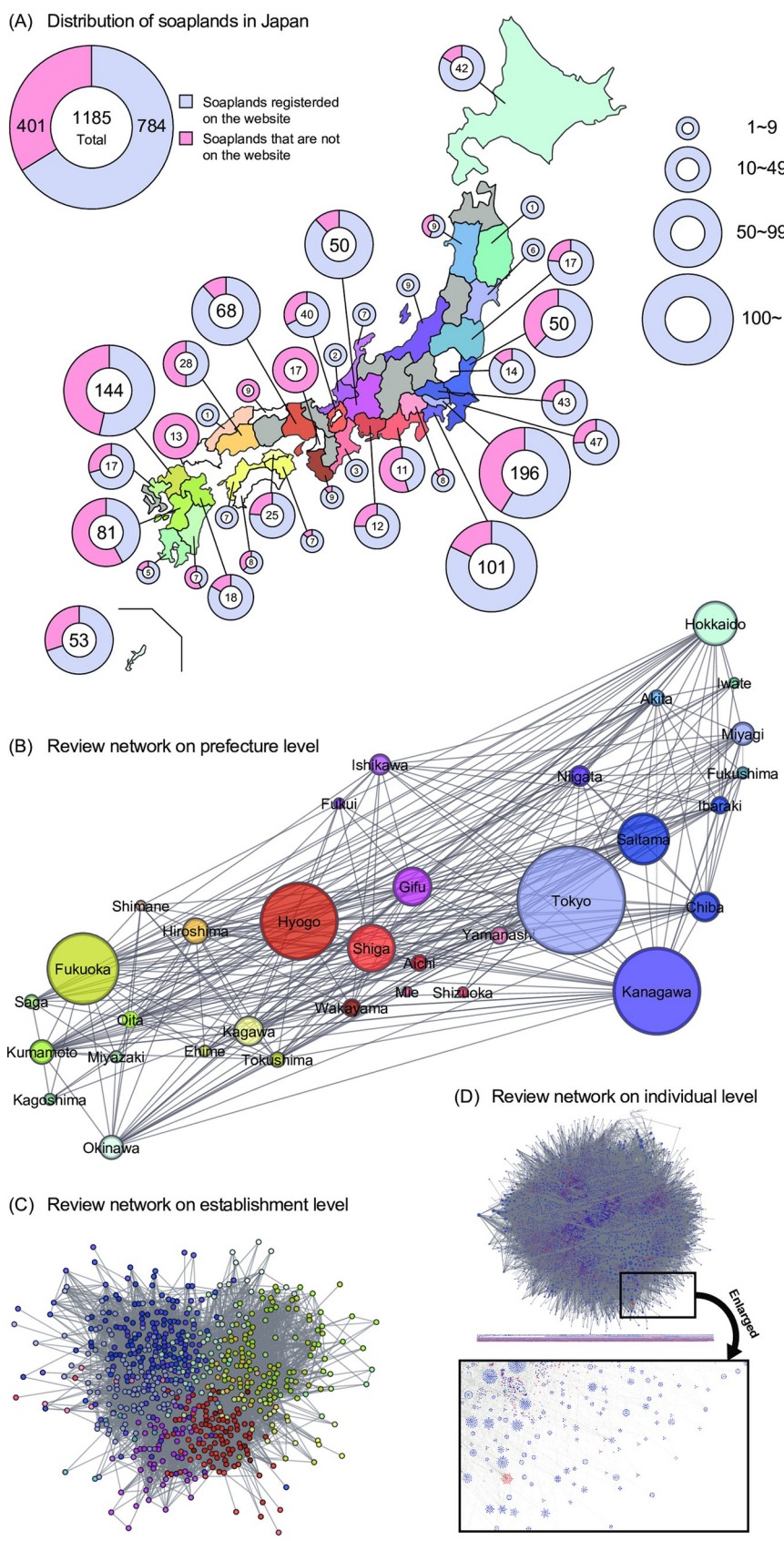

(A) Distribution of soaplands in Japan

1185 Total — 401 / 784

Soaplands registered on the website
Soaplands that are not on the website

1~9
10~49
50~99
100~

(B) Review network on prefecture level

(C) Review network on establishment level

(D) Review network on individual level

**Fig 1. The commercial sex network in Japan.** (A) Distribution of soaplands in Japan and the percentage of establishments registered on the surveyed website. The gray areas show the nine prefectures without soaplands, and the white areas show the four prefectures where review data were not obtained. (B) Review network at the prefecture level. The circles represent the prefectures, and their sizes indicate the number of reviews. The links between prefectures are weighted by the number of MCs who wrote reviews of the FCSWs in the two prefectures, and the weight of the link is defined as the number of such MCs and is illustrated in different thicknesses. (C) Review network at the establishment level. The circles represent the establishments, and their sizes were equal. (D) Bipartite network of FCSWs and MC. Red and blue nodes represent FCSWs and MCs, respectively.

degree (Fig 2B). In Type I, only the degree distributions of FCSWs and MCs are preserved, and other network structures are destroyed. Type II additionally preserves location information in the form of prefectures. In other words, network properties originating from the tendency of MCs to move between prefectures is conserved. Type III additionally preserves information of establishments. This means that the network properties originating from the tendency of MCs to use certain establishments are conserved. Thus, Types II and III are introduced to explore the influence of prefectures and establishments, respectively.

Fig 3A shows the cumulative distributions of degrees. As a result of model selection based on Bayesian information criteria (BIC) and Akaike's information criteria (AIC), the power-law distribution was the best fit (S1 Table). The network exhibited large giant connected components (GCCs), which included 62,917 nodes (FCSWs and MCs). Therefore, 89.0% of the nodes could be connected to each other through the review network. The average path length of the GCC was 9.87. We compared this to the artificial networks created by swapping links (Fig 3B). The average path length of the actual network is close to that of Type III artificial networks. Even if MCs make random visits throughout the country (i.e., Type I), the average resulting path length would only be three-quarters. We calculated the bipartite clustering coefficient proposed by Robins and Alexander [18], which was 0.028. The conventional clustering coefficient is defined as the ratio of 3-cycles and 2-paths, while the bipartite clustering coefficient is defined as the ratio of 4-cycles and 3-paths (see Material and Methods). This value is

**Table 1. Characteristics of a sexual contact network extracted by reviewing a sex industry website.**

| | | |
|---|---|---|
| **Organization of the website** | Number of registered establishments | **784** |
| | Number of establishments listing individual reviews | 684 |
| | Total number of registered FCSWs | 38,964 |
| **Size of the bipartite network** | Total number of reviewed FCSWs | 17,341 |
| | Total number of reviewing MCs | 55,824 |
| | Number of links excluding duplicates (number in parentheses includes duplicate) | 89,543 (121,988) |
| **Structural indicators of the bipartite network** | Size of giant connected components (GCC) | 62,917 (86.0%) |
| | Second largest component | 18 |
| | Average path length (in GCC) | 9.87 |
| | Diameter (in GCC) | 33 |
| | Bipartite clustering coefficient | 0.028 |
| **Degree distribution of FCSWs** | Power exponent $\alpha$ | 2.61 |
| | $k_{\min}$ | 10 |
| **Degree distribution of MCs** | Power exponent $\alpha$ | 3.16 |
| | $k_{\min}$ | 5 |
| **Degree correlation (assortativity coefficient)** | Between FCSWs and MCs | −0.181 |
| | Among FCSWs | 0.524 |
| | Among MCs | 0.150 |

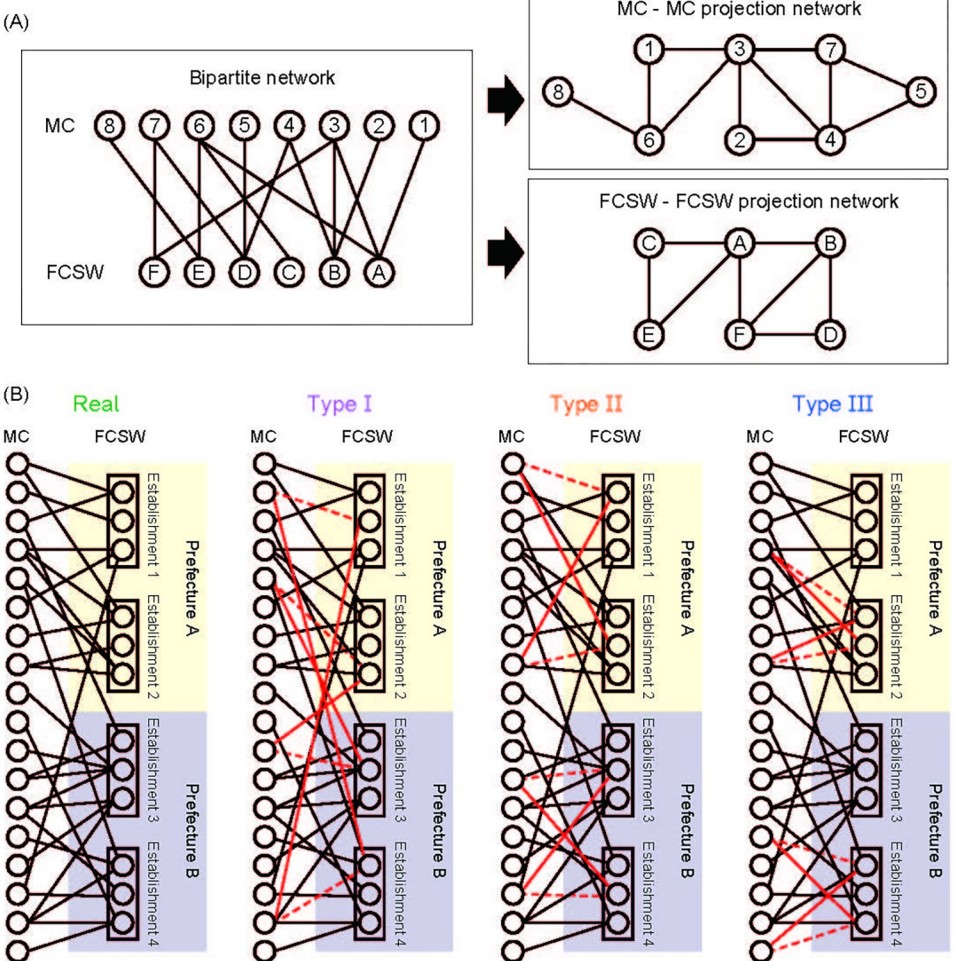

**Fig 2. Bipartite network and projection networks; artificially generated networks; average path length and bipartite clustering coefficient.** (A) Bipartite network of FCSWs and MCs. A link implies that there is at least one review between the FCSW and MC. For the MC–MC projection network, a link implies that the two MCs reviewed at least one common FCSW. For the FCSW–FCSW projection network, a link implies that two FCSWs were reviewed by at least one common MC. (B) For comparison, we used three types of artificially generated networks. For each case, we generated 100 networks where a sufficient number of links were rewired.

significantly larger than in any artificial network (Fig 3C). These results show that the observed network maintains a small-world property.

We calculated the assortativity coefficient to check degree correlation (see Material and Methods). The assortativity coefficient for the bipartite network FCSWs and MCs is −0.18. This negative correlation indicates that popular FCSWs are shared by many infrequent MCs, as was observed in Brazil [11] and the UK [16]. In contrast, the assortativity coefficients for MC–MC and FCSW–FCSW projection networks are 0.524 and 0.150, respectively. In these cases, the assortativity coefficients were positive, suggesting that active MCs tend to review common FCSWs, and popular FCSWs tend to be reviewed by common MCs. Fig 3D–3F show that the above trends are significant. Since the degree correlation cannot be determined from the assortativity coefficient alone [2], we also calculated the average degree of neighbors of degree $k$ nodes, and the above results were supported as shown in S2 Fig.

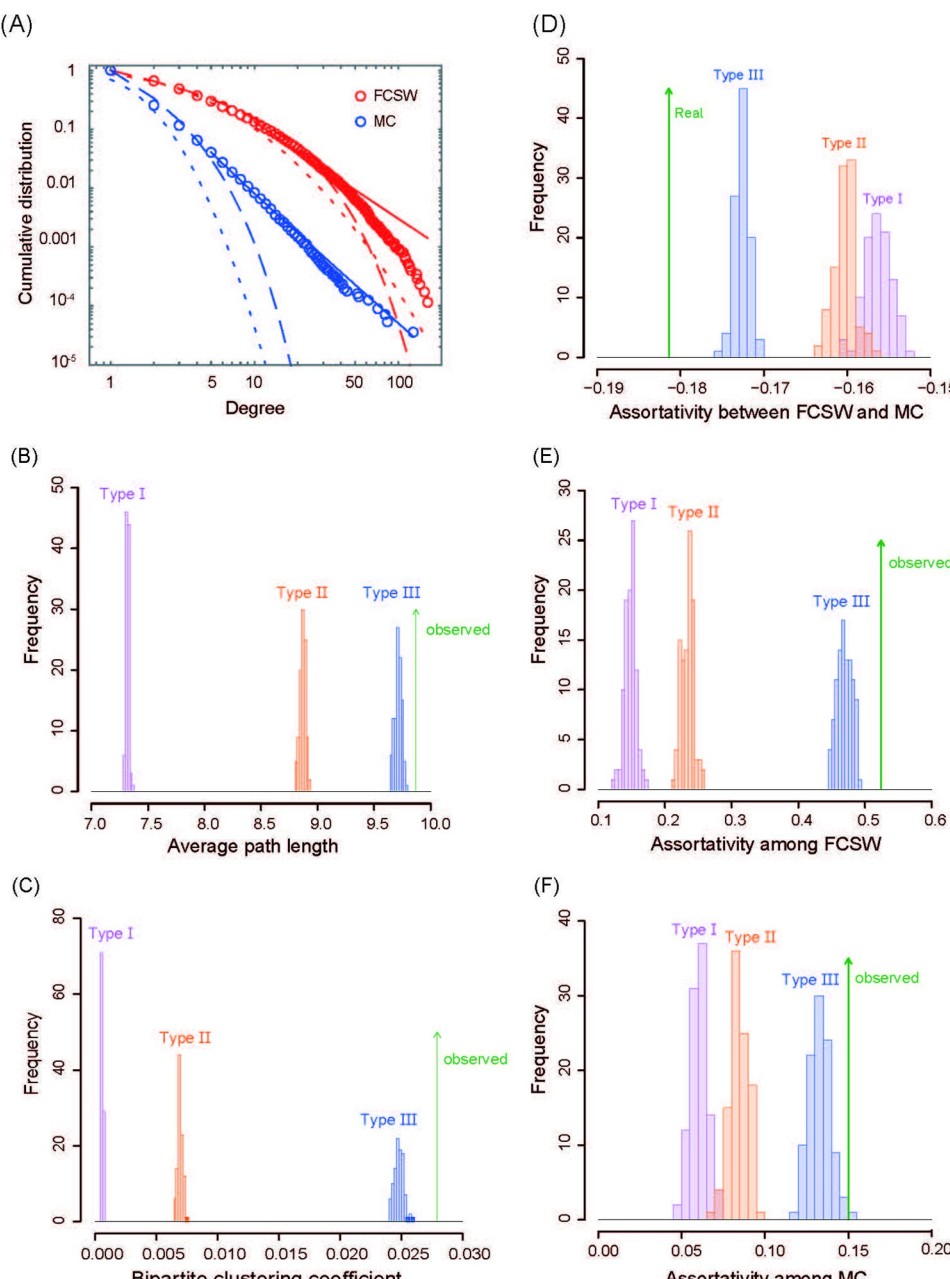

**Fig 3. Results of network analysis.** (A) Cumulative distribution of degree. Circles represent real data. Red and blue correspond to FCSW and MC, respectively. The solid lines, dashed curves, and dotted curves correspond to power-law distribution, negative binomial distribution, and log-normal distribution, respectively. The best fit can be evaluated by BIC or AIC, as shown in S1 Table. (B) Aaverage path lengths. (C) Bipartite clustering coefficients. (D), (E), (F) correspond to the assortativity coefficient between FCSWs and MCs, that for FCSW–FCSW projection networks, and that for MC–MC projection networks, respectively. The green upward arrows represent the assortativity coefficients of the observed network. For comparison, we show the result for three types of artificially generated networks, where 100 networks are prepared and their distributions are shown.

To evaluate the robustness and fragility of the network, we investigated the changes in the three network indices when nodes were deleted from a high degree. Fig 4A shows the change in the size of the GCC; the red and blue curves depict cases when FCSWs and MCs are removed from a high degree, respectively. The horizontal axis is the proportion of removed

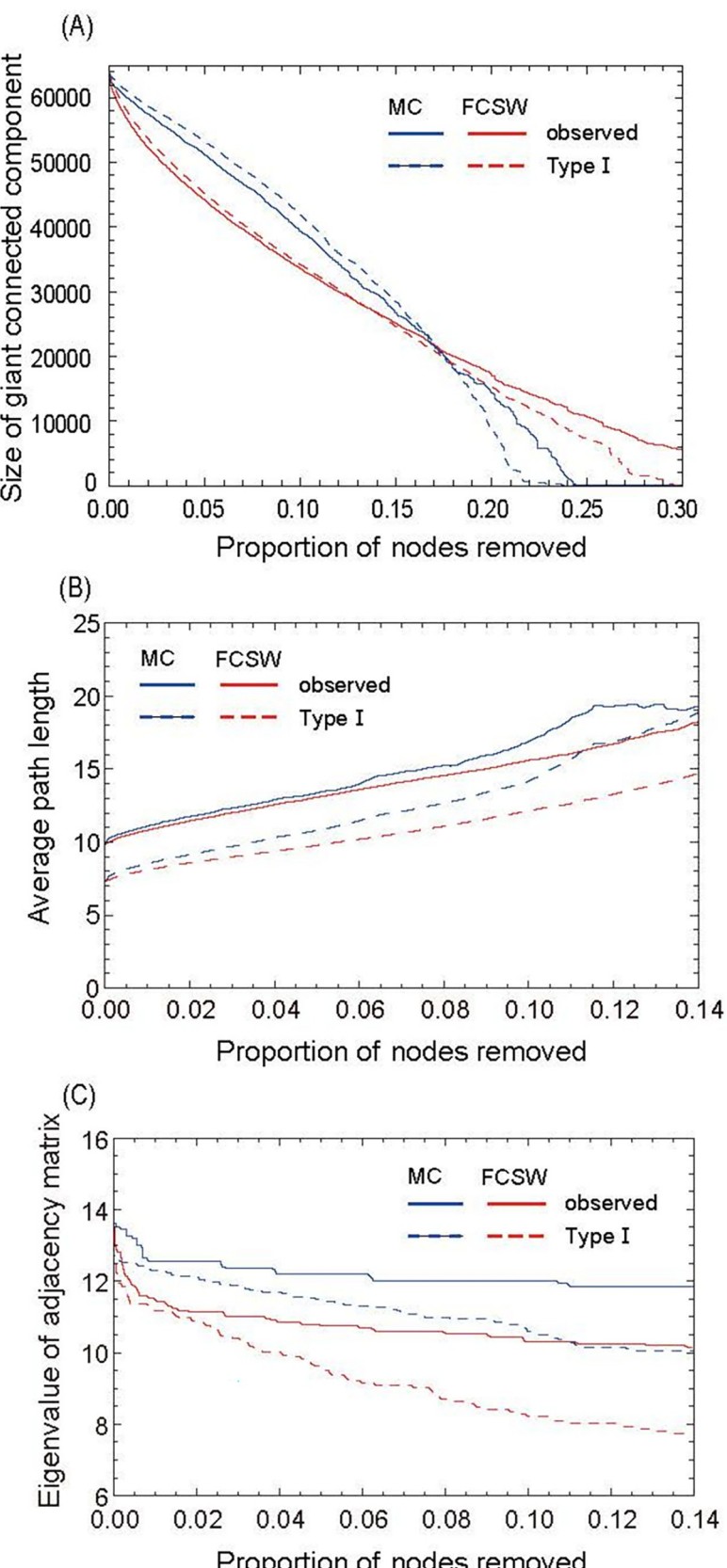

**Fig 4. Robustness to selective nodes removal by degree.** The horizontal axis indicates the proportion of nodes removed, and the vertical axis shows the (A) size of GCC, (B) average path length, and (C) maximum eigenvalue of the adjacency matrix, which is related to the infectious disease model in the network (see Materials and Methods). The solid and dashed curves show the results for the observed network and the Type I artificial network, respectively. Red and blue correspond to FCSW and MC, respectively.

nodes, and the solid and dashed curves denote the cases with observed networks and those with Type I artificial networks, respectively. Cases with Type II and III networks behave somewhere between these two cases (omitted in the figure). Removing popular FCSWs had a more rapid impact on both observed and artificial networks. Removing a sufficiently large percentage of MCs was more effective, but because MCs outnumbered FCSWs, the impact of removing the most popular FCSWs was highly significant. Fig 4B shows the average path length when nodes were deleted, such that there was a sufficiently large GCC. However, the average path length increased faster when MCs were removed. Fig 4C shows the maximum eigenvalue of the adjacency matrix ($\Lambda_1$), which implies the strength of the spread of sexually transmitted infections (STIs) when a simple epidemic model is considered (Materials and Methods). For the observed network, the maximum eigenvalue decreased faster when FCSWs were removed.

## Discussion

We used customer reviews to analyze human sexual contact networks in Japan. We compared three artificially generated networks via the random rewiring of links (Fig 2B) to the observed network. The sexual contact networks investigated exhibited a small-world property with short average path lengths and large clustering coefficients, but were heavily influenced by establishments and locations (prefectures). The clustering coefficients for the Type III network were very close to the real values (Fig 3C). This can be attributed to the fact that an MC tends to visit more than one FCSW within the same establishment. Thus, the establishment is an important factor in the MC's choice of FCSW. The average path length for Type II networks was not minor (Fig 3B). This shows the importance of crossing prefectures, implying that the migration patterns of MCs have a large impact on the network structure.

A negative correlation was observed in the degrees between FCSWs and MCs, as in previous studies [11,16] (Fig 4A and 4B). Popular FCSWs tend to be associated with many low-frequency MCs, and regular MCs tend to connect with less popular FCSWs. For FCSWs, gaining popularity among low-frequency MCs may be beneficial. We also constructed FCSW–FCSW and MC–MC projection networks based on MCs and FCSWs, respectively (Fig 2A). These networks showed a significant positive correlation in degrees (Fig 4C and 4D), i.e., popular and less popular FCSWs tend to only connect with popular and less popular FCSWs, respectively. Similarly, high- and low-frequency MCs tend to connect only with high- and low-frequency MCs, respectively. If this finding is true for actual sexual networks, it could be important for STIs that only affect one sex; hence, it requires future study.

Nodes were removed to evaluate the robustness of the sexual networks. When the nodes were removed from a higher degree, the network structure change was more complex than expected. FCSW and MC removal exhibited a larger effect on the size of the GCC and the average path length, respectively (Fig 4A and 4B). We also investigated the maximum eigenvalues of the adjacency matrix as a more rigorous measure, considering infection dynamics. The results showed that FCSW removal was more effective (Fig 4C). Comparison with the result for Type I network suggests that MC removal has only small effect if the removal is within approximately 1%. Therefore, infection control measures focused on FCSWs may be considered more effective in deterring infections. However, the average path length decreased faster when MCs were removed; thus, the role of MCs should not be underestimated. Active MCs

with a large number of contacts with relatively inactive FCSWs can create shortcuts that bridge various establishments and distant areas, i.e., active MCs can fulfill a function that Granovetter [19] called the "strength of weak ties."

We emphasize that our study should not give the impression that soaplands (and many other types of sex trade industry in Japan) are possible hotspots of STIs. In fact, our analysis does not provide any evidence or relations between STIs and sex industry; furthermore, a previous study reported that the incidence of STIs among FCSWs in soaplands was relatively low [20]. However, if our study were to be applied to public health, the following discussion could be derived. Some previous studies suggested that human sexual networks are highly heterogeneous, with many people having few sexual partners, and a few active people having many sexual partners (i.e., scale-free property), which implies that STIs and other infectious diseases spread easily [7–10]. Thus, popular FCSWs and active MCs (those with a high degree) can have a significant impact on the entire network, possibly spreading STIs nationwide. This is because active FCSWs constitute an important pathway of infection propagation in commercial sex networks, and MCs play an essential role as weak ties. Therefore, our results reveal the potential STI diffusion effects of active MCs who use multiple brothels in different areas, which has been claimed by some previous studies [21–23].

One major difference between the current study and previous studies of online-sex- worker networks [11–16] is that the establishments exist independently of the website. We were able to use three types of randomized networks to see the effect of the establishments and their locations. Although we did not collect the attribute data of FCWSs and MCs, discussion could be furthered by using these data. In particular, assigning MCs to prefectures in which they primarily use sex establishments, may allow for a more accurate analysis of the effect of MC migration. Analysis using the attribute values will be the subject of future work.

It should be noted that the contacts revealed by reviews are only a small fraction of the actual commercial sexual contacts. According to the website, the number of members exceeded 2 million in March 2021, but only less than 3% of them wrote reviews. Another important limitation is that MCs can have more than one accounts with their linked mobile phone numbers, and FCSWs can have more than one account if they work in more than one establishments. This is an unavoidable limitation of the study since MCs and FCSWs are anonymized. We did not consider the temporal order of the reviews, but that may be important as shown in [12]. Thus, the network presented here is only a small part of the long history of prostitution in Japan. However, the observed network is much larger and more well controlled than those previously examined, and the results are expected to be more reliable.

## Materials and methods

### Study website

In this study, we focused on one of the largest adult entertainment websites in Japan. The targeted website is available only in Japanese, and it leverages a reservation system to book engagements with FCSWs. This website is open to anyone aged 18 years or older, and the reviews can be read by anyone. This website began its review service in November 2018. Although many similar review sites do not have reliable sources of information, the target website does. We have selected the largest website among those with reliable sources for this study. Whether an establishment is registered on this site is the manager's decision. At the time of our survey, there were 784 soaplands registered on this website. As shown in Fig 1A, this website covers more than half of the soaplands in Japan. At the time of our survey, the website included information on 38,964 FCSWs.

An MC can become a member for free by registering a mobile phone number, and a member can post reviews using a registered handle. According to the website, the number of members exceeded 2 million in March 2021. In contrast, establishments pay a registration fee to the website management company and update information about their staff (FCSWs) for promotion. The information on the website contains establishment locations, service fees, and features (e.g., portrait, age, and measurements) of FCSWs. Each review includes the name of the FCSW and the alias of the reviewer. Reviews are written voluntarily, but some soaplands offer incentives in the form of discount coupons. If the establishment or website operator deems the content of the review inappropriate (e.g., false and slanderous), the review may not be published or may be deleted.

## Data collection

The data were collected using code written in Python, from which we obtained (i) establishment names, (ii) locations (prefectures), (iii) FCSW names, and (iv) reviewer names per review. Data collection was conducted on April 4, 2021 (see S1 Fig). The names of FCSWs and MCs were replaced by arbitrary numbers (S1 Data). We used only those reviews in which both the staff (FCSW) and customer (MC) could be distinguished. Because the information on the website is organized by prefecture, scraping was done by prefecture. No soapland exists in 9 prefectures (Aomori, Gunma, Kyoto, Nagano, Nagasaki, Nara, Okayama, Toyama, and Yamagata) of Japan's 47 prefectures owing to the ordinances of those prefectures. Data for two prefectures (Kochi and Tochigi) on the website could not be obtained owing to network problems on the day of the survey.

## Privacy and ethical issues

In this research, we analyzed a network of publicly available customer reviews on a sex industry website. In other words, we did not deal with data on real contact between persons, but instead with data on connections in virtual space. The website data we collected are open to the public and can be viewed by anyone without any registration. Reviews are published under the MC's alias and their personal information is protected by the website companies. Therefore, the MC's personal information is not accessible by the public. The customer reviews are only published with the informed consent of the FCSWs via the establishments. Note that in Japan, establishments cannot make excessive demands on FCSWs because if an FCSW reports the happenings in the establishment to the police, the establishment will be shut down and the managers imprisoned. When FCSWs quit an establishment, information about them is no longer accessible.

We collected publicly available information in compliance with the terms of the data source, and assigned each user account, which was also anonymous, a randomly chosen number as our ID. Furthermore, we extracted the connections only, and no personal information was included. Note also that the study website was designed to be anonymous from the outset, making it impossible for visitors to the website to identify any FCSWs or MCs. Therefore, there is no risk that any person's privacy will be violated by this research.

In summary, this survey is based exclusively on publicly available data and includes no personal information of human subjects; it does not fall under the category of human subject research. Further, we received a formal letter acknowledging that this study is not categorized as research involving human subjects, and therefore does not require authorization, from the Human Research Ethics Committee (Chair, Toshihiko Dozono) of Shizuoka University, to which the corresponding author belongs.

## Network analysis

We performed a network analysis as follows. We focused on a bipartite network, that is, the nodes were divided into two groups (FCSWs and MCs), and there were no links between the same groups.

**Degree distribution.**   Degree distributions are the most commonly used metrics in network analysis. Because the networks are bipartite, we must consider two different degree distributions: the degree of an MC is defined as the number of FCSWs reviewed, and the degree of an FCSW is defined as the number of MCs who reviewed the FCSW. We checked whether these distributions had a power-law tail:

$$P(k) \propto k^{-\alpha} \text{ for } k \geq k_{\min}. \tag{1}$$

The parameters $\alpha$ and $k_{\min}$ are estimated by model selection using AIC and BIC. Furthermore, the model selection confirmed that the power distribution was better than negative binomial and lognormal distributions [10,24,25].

**Average path length.**   To examine whether the network could be classified as small-world, we calculated the average path length, which is the average number of steps along the shortest paths for all pairs of network nodes. If the average path length is small, the network is small-world.

**Clustering coefficient.**   The clustering coefficient measures the closeness of a node's neighbors [4]. Review networks are bipartite; therefore, traditional clustering coefficients cannot be used as such network structures do not produce triangles. Several clustering coefficients have been proposed for bipartite networks [18,26]. Here, we used the bipartite clustering coefficient proposed by Robins and Alexander, which is the ratio between the number of four 4-cycles and the number of 3-paths [18]:

$$C_{RA} = \frac{4C_4}{L_3},$$

where $C_4$ is the number of 4-cycles and $L_3$ is the number of 3-paths in a bipartite network. If the bipartite clustering coefficient is high, then multiple FCSWs with an MC tend to be with another MC, and multiple MCs with an FCSW tend to be with another FCSW. For calculating the bipartite clustering coefficient, we used *reinforcement_tm* from the tnet package in R.

**Degree correlation.**   Degree correlation is an important indicator in network analysis. To quantify the degree of correlation, we calculated the assortativity coefficient, which is the Pearson correlation coefficient of the degree between pairs of linked nodes [27]:

$$r = \frac{\sum_{ij}\left(A_{ij} - \frac{k_i k_j}{2m}\right)k_i k_j}{\sum_{ij}\left(k_i \delta_{ij} - \frac{k_i k_j}{2m}\right)k_i k_j},$$

where $k_i$ is the degree of individual $i$, and $m$ is the number of links. If the degree correlation between FCSWs and MCs is positive (negative), it indicates that MCs who frequently visit soaplands are more (less) likely to be associated with popular FCSWs. Similarly, we examined FCSW–FCSW and MC–MC projection networks (Fig 2A), which are networks in which FCSWs are connected through MCs and those of MCs connected through FCSWs, respectively. By examining these projection networks, we can see the connections among FCSWs or among MCs. To ascertain the details of the dependence of degree correlation we also measured the average degree of neighbors of degree-$k$ nodes and studied its $k$-dependence (S1 Fig).

**Robustness of the network.**   To understand the spreading phenomenon on the network, we examined the robustness of the network against node removal. It is known that in scale-

free networks, deleting nodes starting from those with a higher degree makes the network fragile. Here, we compared the case of removing FCSWs from higher-degree nodes with the case of removing MCs from higher-degree nodes. To measure robustness, we used three indicators: GCC size, average path length, and maximum eigenvalue ($\Lambda_1$) of the adjacency matrix ($A_{ij}$). $\Lambda_1$ is proportional to infectivity, considering a simple epidemiological model in which the infection propagates along the unweighted links of the network. For example, in an Susceptible-infected-susceptible (SIS) model, if the probability of infection of individual $i$ is denoted by $\rho_i$, $\beta$ is the infection rate, and $\gamma$ is the cure rate, we obtain

$$\frac{d\rho_i}{dt} = -\gamma\rho_i + (1 - \rho_i)\beta\sum_{j=1}^{N}A_{ij}\rho_j$$

using individual-based mean-field theory [28–31]. For the effective transmission rate $\lambda = \beta/\gamma$, the infection threshold is given as

$$\lambda \geq \lambda_c, \lambda_c = \frac{1}{\Lambda_1}.$$

An exact calculation [32] yields

$$\lambda_c \geq \frac{1}{\Lambda_1}.$$

The same result was obtained when SIR, SIRS, or SEIR was used instead of SIS [33]. Note that the weights of the links in the network and the temporal dynamics of the contacts are not considered here. Thus, $\Lambda_1$ is an indicator of the strength of spread on a simple graph, not an indicator that can predict the final number of infected persons.

**Randomly rewired networks.**    To verify the implications of the obtained results, we compared them to cases from artificially generated networks in which the links were randomly rewired. We used three types of artificially generated networks, including Type I, which preserves the degree of all nodes; Type II, which does not change prefectures or degrees; and Type III, which does not change establishments or degrees (Fig 2B). By comparing these three artificially generated networks, we indirectly validated the influence of prefectures and establishments. For each case, we generated 100 networks in which a sufficient number of links were rewired.

## Supporting information

**S1 Data. List of reviews.**
(CSV)

**S1 Fig. The total number of reviews for each month.**
(DOCX)

**S2 Fig. Degree correlation curve.** Plot of the average degree of the neighborhood of degree k nodes as a function of k. The green marks correspond to the observed networks, whereas the purple ones correspond to the randomized network preserving the degree distribution. (A) and (B) show the average degree of neighbors of degree k nodes for FCSWs and MCs, respectively. The assortativity coefficient is -0.181. This negative correlation indicates that popular FCSWs are shared by many infrequent MCs. (C) and (D) show the same analysis for the FCSW–FCSW and MC–MC projection networks (right panel in Fig 2A). The assortativity coefficients were positive, suggesting that active MCs tend to review common FCSWs, and

popular FCSWs tend to be reviewed by common MCs.
(DOCX)

**S1 Table. Model selection for degree distribution.**
(DOCX)

**S1 Appendix. Soaplands in Japan.**
(DOCX)

## Acknowledgments

We are grateful to Dr. Jin Yoshimura for his valuable comments on this paper.

## Author Contributions

**Conceptualization:** Hiromu Ito, Taro Yamamoto, Satoru Morita.

**Data curation:** Satoru Morita.

**Formal analysis:** Satoru Morita.

**Funding acquisition:** Hiromu Ito, Satoru Morita.

**Investigation:** Satoru Morita.

**Methodology:** Hiromu Ito, Keiko Shigeta, Satoru Morita.

**Project administration:** Taro Yamamoto, Satoru Morita.

**Resources:** Keiko Shigeta.

**Software:** Keiko Shigeta.

**Supervision:** Taro Yamamoto, Satoru Morita.

**Validation:** Satoru Morita.

**Visualization:** Hiromu Ito, Satoru Morita.

**Writing – original draft:** Hiromu Ito, Satoru Morita.

**Writing – review & editing:** Hiromu Ito, Taro Yamamoto, Satoru Morita.

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
