## [Decision Letter · Decision Letter 0]

21 Jun 2022

PONE-D-22-07615Exploring sexual contact networks by analyzing a nationwide commercial-sex review websitePLOS ONE

Dear Dr. Morita,

Thank you for submitting your manuscript to PLOS ONE. After careful consideration, we feel that it has merit but does not fully meet PLOS ONE’s publication criteria as it currently stands. Therefore, we invite you to submit a revised version of the manuscript that addresses the points raised during the review process.

We look forward to receiving your revised manuscript.

Kind regards,

Hocine Cherifi

Academic Editor

PLOS ONE

Journal Requirements:

“This work was partially supported by the Japan Society for the Promotion of Science (JSPS) KAKENHI (nos. 17H04731, 19KK0262, 21H01575 to H.I.; JP19H05731 to T.Y.; and 18K03453, 21K03387 to S.M.). The funders have/had no role in the study design, data collection and analysis, decision to blish, or preparation of the manuscript.”

Reviewers' comments:

Reviewer's Responses to Questions

**Comments to the Author**

1. Is the manuscript technically sound, and do the data support the conclusions?

Reviewer #1: No

Reviewer #2: Yes

2. Has the statistical analysis been performed appropriately and rigorously? 

Reviewer #1: No

Reviewer #2: Yes

3. Have the authors made all data underlying the findings in their manuscript fully available?

Reviewer #1: Yes

Reviewer #2: Yes

4. Is the manuscript presented in an intelligible fashion and written in standard English?

Reviewer #1: No

Reviewer #2: Yes

5. Review Comments to the Author

Reviewer #1: The study attempts to study networks of sexual contacts in Japan derived from online reviews.

The manuscript presents an interesting data set and some interesting results however misses to provide robust findings and discuss implications of the network analysis to public health, or sociology, or economy, limiting itself to a network data analysis study.

Furthermore, it needs a thorough review to improve the language, explanations, remove redundant information, and clarify the purpose and meaning of some measures.

It also needs a more proper contextualisation with the literature given that several studies have addressed topics discussed in the manuscript, such as Internet-mediated prostitution and epidemic spreading.

I cannot recommend the current version of the manuscript for publication on PLoS ONE or any other journal but I encourage the authors to address the comments below and revise the manuscript accordingly.

- The meaning/scope of Prefecture should be explained, given that this definition is not much used in most countries.

- An explanation of "soapland" should be given already in the introduction, e.g. the text in the methods could be more concise and merged in the introduction.

- Female commercial sex workers -> Female sex workers

- p4, l61-65. The authors should write these points using a more public-health language, i.e. that sex-workers are a risk group or at risk group, that is more exposed to potential STIs in case of unprotected sex, etc... similarly to clients.

- The 3 types of random models should be better explained. What do authors mean by "randomly rewired networks without changing the XX"? I assume that some network properties are being conserved but it is not clear.

-- fig 2b should first show the empirical network and then the randomized models

- The purpose of fig 1C and fig 1D must be clarified. In principle, they do not bring any relevant insights to understand the follow up analysis. I suggest only showing the "enlarged" part of fig 1d.

-- Similarly, fig 1A could be simplified.

- Table 1 should be simplified and only contain relevant information that brings insights.

- The bi-partite clustering coefficient should be described since it is not a standard network measure.

- Fig 4 should be removed since it is not much informative, given that assortativity can be measured and conveys the same information.

- P12, l250. Which year and/or month was the data collected? It corresponds to which period? Part of this is written elsewhere.

- The section "Methods: Spreading on the Network" is misleading because there is no epidemic model being studied in this manuscript. Eq 2 is not necessarily true for other models than SIS and SIR, and the authors should specify what they want to study/capture with this epidemic threshold? It's OK to use this measure but limitations should be clearly described, and the scope in terms of STIs clarified.

- The text in Methods should (in general) be more concise and show the definitions of measures and equations when appropriate (e.g. clustering coefficient for bi-partite network), the randomisation procedure appears in the methods and in the main body of the manuscript (unclear in both cases).

- What are the implications of these findings for public health (or other disciplines)? How do they compare with previous literature?

- I recommend the authors to apply other strategies to study the robustness of the network, e.g. considering specific prefectures or soapland establishments, or other network measures at the individual level. That would enrich substantially the analysis and could provide several insights.

- Some relevant papers that would help authors to contextualise their findings:

--- about online prostitution data/reviews & advertisement:

https://link.springer.com/article/10.1007/s42001-021-00156-2

https://www.oxfordhandbooks.com/view/10.1093/oxfordhb/9780199915248.001.0001/oxfordhb-9780199915248-e-17

https://www.oxfordhandbooks.com/view/10.1093/oxfordhb/9780199915248.001.0001/oxfordhb-9780199915248-e-3

--- about epidemic spreading on sexual networks of sex-workers and their clients.

https://journals.plos.org/ploscompbiol/article?id=10.1371/journal.pcbi.1001109

https://pubmed.ncbi.nlm.nih.gov/12650777/

--- about sampling sensitive data

https://academic.oup.com/comnet/article-abstract/9/6/cnab034/6420764

Reviewer #2: Thank you very much for your interesting article. You were scraping data from an online website for soaplands and used information on sex workers and male clients to construct a social network and analyse the properties of this network, also with regard to the spread of a hypothetical STI. The article is well written and easily understandable. I have some minor comments, which in my opinion would strengthen the understanding of the data and underline the contribution of your work to understanding the spread of STIs.

- You state that 55% of soaplands are covered by the website. From the Map A in Figure 1, I think that there is some geographical bias with regard to soaplands missing on the website. Could this be due to soaplands of specific regions using a different website? (Or in other terms, how random are soaplands missing on the website?)

- Are there any estimations on the share of commercial sex contacts outside of soaplands? I.e., what role do soaplands play in commercial sex? Are sexual contacts in soaplands more, equal or less likely to perform safer sex? You state that there is no academic investigation in the discussion section. But is there any information on legal proceedings for illegal sex work?

- A similar question would be, if you could provide some context of the incidences of STIs in Japan. Is there any information where people get infected with STIs? Which role deos commercial sex play as an infection setting compared to other settings (private contacts, injecting drugs, etc.).

- Would it not be possible to obtain data for the prefectures Kochi and Tochigi now? I get that it would preferrable to have the data collection happening on the same day for all, but IMHO it would be better to have a more complete dataset from different dates than an incomplete dataset from one day. (somewhat related: you state that Kochigi does not allow soaplands, so would it matter that data was not accessible?)

- Is there some information on the website how many MCs are registered overall?

- Can you do some sensitivity analyses how MCs and FCSWs with no reviews could affect the network properties if they behaved differently than the MCs and FCSWs with reviews?

- Is there a way to avoid having duplicates of FCSWs if they work in different soaplands by trying to find duplicates by name or provided features (potrait, age, etc.)?

6. PLOS authors have the option to publish the peer review history of their article (what does this mean?). If published, this will include your full peer review and any attached files.

Reviewer #1: No

Reviewer #2: **Yes: **Stefan Scholz

---

## [Decision Letter · Decision Letter 1]

18 Oct 2022

Exploring sexual contact networks by analyzing a nationwide commercial-sex review website

PONE-D-22-07615R1

Dear Dr. Morita,

We’re pleased to inform you that your manuscript has been judged scientifically suitable for publication and will be formally accepted for publication once it meets all outstanding technical requirements.

Kind regards,

Hocine Cherifi

Academic Editor

PLOS ONE

Additional Editor Comments (optional):

The paper is now ready for publication. Congratulations for a nice worK.

Reviewers' comments:

Reviewer's Responses to Questions

**Comments to the Author**

1. If the authors have adequately addressed your comments raised in a previous round of review and you feel that this manuscript is now acceptable for publication, you may indicate that here to bypass the “Comments to the Author” section, enter your conflict of interest statement in the “Confidential to Editor” section, and submit your "Accept" recommendation.

Reviewer #1: All comments have been addressed

2. Is the manuscript technically sound, and do the data support the conclusions?

Reviewer #1: Yes

3. Has the statistical analysis been performed appropriately and rigorously? 

Reviewer #1: Yes

4. Have the authors made all data underlying the findings in their manuscript fully available?

Reviewer #1: No

5. Is the manuscript presented in an intelligible fashion and written in standard English?

Reviewer #1: Yes

6. Review Comments to the Author

Reviewer #1: I thank the authors for addressing most of my comments satisfactorily. I highlight that some points could be discussed further but I acknowledge the efforts and now recommend the manuscript for publication on PLoS ONE.

As a minor note, I was asked by the editorial office whether the data set used in this study is being provided to comply with the PLoS data policy. I did not find it in the submission or a clear link/direction on how to get it. Maybe I missed it, but please, clarify how the data can be accessed.

7. PLOS authors have the option to publish the peer review history of their article (what does this mean?). If published, this will include your full peer review and any attached files.

Reviewer #1: No

---

## [Editor Report · Acceptance letter]

24 Oct 2022

PONE-D-22-07615R1 

Exploring sexual contact networks by analyzing a nationwide commercial-sex review websiteExploring sexual contact networks by analyzing a nationwide commercial-sex review website 

Dear Dr. Morita:

I'm pleased to inform you that your manuscript has been deemed suitable for publication in PLOS ONE. Congratulations! Your manuscript is now with our production department. 

Kind regards, 

on behalf of

Professor Hocine Cherifi 

Academic Editor

PLOS ONE